# Rest-task modulation of fMRI-derived global signal topography is mediated by transient coactivation patterns

**Jianfeng Zhang** [1,2]*, **Zirui Huang**[3], **Shankar Tumati**[4], **Georg Northoff**[1,4,5,6]*

1 Mental Health Center, Zhejiang University School of Medicine, Hangzhou, China, 2 College of Biomedical Engineering and Instrument Sciences, Zhejiang University, Hangzhou, China, 3 Center for Consciousness Science, Department of Anesthesiology, University of Michigan Medical School, Ann Arbor, Michigan, United States of America, 4 Institute of Mental Health Research, University of Ottawa, Ottawa, Canada, 5 Center for Cognition and Brain Disorders, Hangzhou Normal University, Hangzhou, China, 6 Graduate Institute of Humanities in Medicine, Taipei Medical University, Taipei, Taiwan

* jianfeng_zhang@zju.edu.cn (JZ); georg.northoff@theroyal.ca (GN)

**Data Availability Statement:** Data are from the Human Connectome Project S1200, which is publicly available at https://www.humanconnectome.org/. Data points underlying figures, together with a readme text file explaining

## Abstract

Recent resting-state functional MRI (fMRI) studies have revealed that the global signal (GS) exhibits a nonuniform spatial distribution across the gray matter. Whether this topography is informative remains largely unknown. We therefore tested rest-task modulation of GS topography by analyzing static GS correlation and dynamic coactivation patterns in a large sample of an fMRI dataset ($n = 837$) from the Human Connectome Project. The GS topography in the resting state and in seven different tasks was first measured by correlating the GS with the local time series (GSCORR). In the resting state, high GSCORR was observed mainly in the primary sensory and motor regions, whereas low GSCORR was seen in the association brain areas. This pattern changed during the seven tasks, with mainly decreased GSCORR in sensorimotor cortex. Importantly, this rest-task modulation of GSCORR could be traced to transient coactivation patterns at the peak period of GS (GS-peak). By comparing the topography of GSCORR and respiration effects, we observed that the topography of respiration mimicked the topography of GS in the resting state, whereas both differed during the task states; because of such partial dissociation, we assume that GSCORR could not be equated with a respiration effect. Finally, rest-task modulation of GS topography could not be exclusively explained by other sources of physiological noise. Together, we here demonstrate the informative nature of GS topography by showing its rest-task modulation, the underlying dynamic coactivation patterns, and its partial dissociation from respiration effects during task states.

## Introduction

One of the major confounds that limits the cognitive and clinical applications in functional MRI (fMRI) is the global signal (GS), which is defined as the spatial average of time-varying blood oxygen level–dependent (BOLD) signals [1,2]. The GS is often considered to represent

the structure of the data, are available on a Dryad repository: https://doi.org/10.5061/dryad.xsj3tx9bw.

**Funding:** This work is supported by the grant from the Ministry of Science and Technology of China, National Key R&D Program of China (2016YFC1306700), the European Union's Horizon 2020 Framework Programme for Research and Innovation under the Specific Grant Agreement No. 785907 (Human Brain Project SGA2), the EJLB-Michael Smith Foundation, the Canada Institute of Health Research (CIHR), the Start-up Research Grant in Hangzhou Normal University to GN, and Zhejiang University Academic Award for Outstanding Doctoral Candidates to JZ. The funders had no role in study design, data collection and analysis, decision to publish, or preparation of the manuscript.

**Competing interests:** The authors have declared that no competing interests exist.

**Abbreviations:** BOLD, blood oxygen level–dependent; CAP, coactivation pattern; CSF, cerebrospinal fluid; DMN, default mode network; fMRI, functional MRI; GS, global signal; GSCORR, GS correlation; HCP, Human Connectome Project; HR, heart rate; ICC, intraclass correlation coefficient; n.s., not significant; ROI, region of interest; rs-fMRI, resting-state fMRI; RVT, respiration volume per time; RVTCORR, RVT correlation; SNR, signal-to-noise ratio; T, tesla.

physiological noise caused by respiratory and cardiac events [3] and has been recommended to be regressed out during data preprocessing prior to secondary analyses.

However, other studies have shown that the GS is not merely nonneuronal noise, and it contains important information about neuronal activity. For instance, the GS in fMRI exhibits a high correlation to electrophysiological measures on the cortical level [4,5]; its global fluctuation may partly stem from subcortical regions (e.g., basal forebrain) relating to arousal [6,7]; and its variance fluctuates across time of day [8]. Finally, abnormalities in the level of the GS have been observed in psychiatric disorders like schizophrenia [9] and bipolar disorder [10]. Together, these findings underscore the potentially informative nature of the GS.

One key feature of the GS is its topographical pattern [6,10–12]. Although the GS is generated by the shared activity across the gray matter, recent studies have demonstrated a nonuniform topographical distribution of GS across brain regions in both monkeys [7] and humans [6,11,13] during the resting state. Higher levels of GS correlation (GSCORR) are observed in the primary sensory regions such as the auditory and visual cortices, and lower levels of GSCORR are seen in the higher-order cortical regions including the prefrontal cortex during the resting state.

The informative nature and thus the functional relevance of resting-state GS topography is still subject to debate. On the one hand, this topography seems to be functionally relevant, as it correlates with cognitive performance in a healthy population [12] and shows abnormal in psychiatric disorders like schizophrenia [11], autism spectrum disorder [14,15], or bipolar disorder [10]. On the other hand, the resting-state GS topography resembles the spatial pattern of a respiration effect [3,16–18]; this raises the question whether the GS topography merely represents physiological noise, i.e., respiration. One way to reconcile the two seemingly contradictory observations seems to be investigating the modulation of GS during task state relative to the resting state and how that stands in relation to respiration effect.

The overarching aim of this study is to probe the informative nature [19] of GS topography by examining its modulation during various cognitive tasks relative to the resting state, i.e., rest-task modulation. For that purpose, we utilize a large sample of an fMRI dataset from the Human Connectome Project (HCP), including 2 days' resting states and seven different tasks. First, we quantified the GS topography and its modulation during different tasks. We sought to demonstrate whether and how external tasks modulate GS topography, e.g., in a task-specific or task-unspecific manner. Second, following a recent study showing how GS may govern the brain's spatiotemporal dynamics [20], we explored the possible mechanism underlying rest-task modulation of GS topography by analyzing its dynamic nature in terms of transient coactivation patterns (CAPs) at the critical time points of GS (i.e., the peaks of GS) [6]. Third, we tested whether rest-task modulation of GS topography on the cortical level stems from the contributions of subcortical regions [6] or can be explained by the nonneuronal physiological noise, such as respiration [3,21]. Finally, given the importance of reproducibility of fMRI studies in general [22], we examined the rest-task modulation in an independent 7 tesla (T) dataset from the HCP with two additional tasks.

## Results

### GS topography in the resting and task states

We first calculated the functional connectivity between the GS and time course in each grayordinate (i.e., gray matter in cortical and subcortical regions) using Pearson correlation (i.e., GSCORR) with Fisher z transformation (Fig 1A). We quantified the GSCORR topography at a group level by averaging the maps of GSCORR across 837 participants for resting states at 2 days and seven tasks, respectively. Consistent with previous findings [6,11], a typical GS

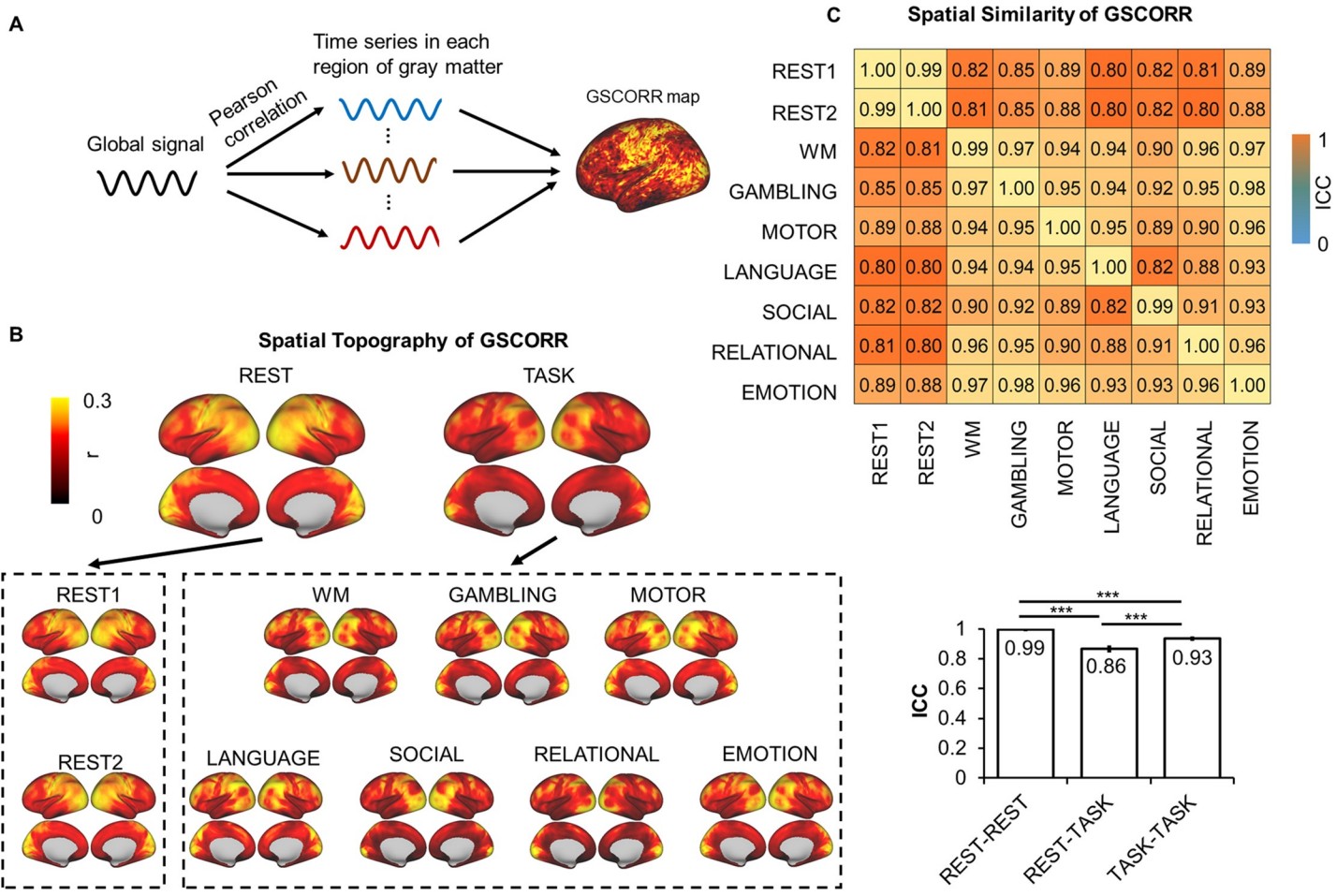

**Fig 1. Spatial topography of global signal correlation.** (A) An illustration of GSCORR as a correlation between global signal and time series in each region of gray matter. (B) Spatial patterns of GSCORR in 2 days' resting state and seven tasks. Top panel yielded the averaged spatial patterns for the resting states and task states, respectively. Bottom panel yielded the spatial pattern in each condition. (C) The spatial similarity was performed by using ICC, at the group level, based on a parcellation with 1,000 ROIs. Top panel demonstrated the ICC matrix between 2 days' resting states and seven tasks. Bottom bar chart demonstrated the ICC ± 95% CI for rest-rest, rest-task, and task-task respectively. ∗∗∗ *p* < 0.001. Data are available at Dryad: https://doi.org/10.5061/dryad.xsj3tx9bw. GSCORR, global signal correlation; ICC, intraclass correlation coefficient; ROI, region of interest; WM, working memory.

topography during resting state was observed, with higher levels of GSCORR in the sensory regions (visual, auditory, and somatosensory regions) and lower levels of GSCORR in the association regions (prefrontal and parietal cortices) (Fig 1B). Interestingly, the GSCORR topography in tasks differed from the resting state, in which higher GSCORR was persisted in visual but not in somatosensory regions (Fig 1B).

We further quantified the spatial similarity of the GS patterns among the resting states and tasks using intraclass correlation coefficient (ICC) [23], which was widely used for investigating the spatial similarity in test-retest reliability studies in resting-state fMRI (rs-fMRI) [22,24,25]. In reporting these findings, we categorized the ICC into five common intervals [26]: 0 < ICC ≤ 0.2 (slight); 0.2 < ICC ≤ 0.4 (fair); 0.4 < ICC ≤ 0.6 (moderate); 0.6 < ICC ≤ 0.8 (substantial); and 0.8 < ICC ≤ 1.0 (almost perfect). Additionally, given the spatial similarity analyses can be confounded by the local spatial adjacency at the grayordinate level, we thus minimized this potential by performing the spatial similarity analyses based on a parcellation template with 1,000 regions of interest (ROIs) [27]. First, we observed an almost

identical GS topography during the resting states on the 2 days (ICC = 0.9986, 95% CI 0.9985–0.9988) (Fig 1C), suggesting a high reliability of GS topography across time. High ICCs were also observed across different tasks (ICC = 0.9356, 95% CI 0.9296–0.9411) (Fig 1C). The ICC was also high between resting state and tasks (ICC = 0.8666, 95% CI 0.8504–0.8813); however, the value was lower compared with both rest-rest and task-task similarities ($p < 0.001$ in both, Fisher z test, Fig 1C). Together, the spatial pattern of GSCORR showed small but significant changes from resting state to task state.

## Task modulation of GS topography

An approach to specify the informative nature of GS topography is rest-task modulation. If GSCORR changes from rest to task, the GSCORR must be assumed to contain some information. For that purpose, we first investigated the overall GSCORR across time (REST1 and REST2) and state (rest and seven tasks). The comparison of GSCORR within the two rests showed no significant difference despite the large sample size ($n = 837$). In contrast, the comparisons of GSCORR between rest and tasks showed a consistent reduction of GSCORR in all tasks (in all contrasts between rest and tasks, t > 12, Cohen's d > 0.4, Bonferroni corrected at $\alpha < 0.01$) (Fig 2A).

The reduction of GSCORR can occur uniformly across all regions in the brain or, alternatively, can be manifested in a nonuniform way across different regions. To address this issue, we performed grayordinate-based comparison between rest and tasks. As the sample size was large and therefore easily reached significance, we used a strict threshold at Bonferroni-corrected $\alpha < 0.01$ for further analyses, and the results were illustrated by Cohen's d as an index of effect size, which is insensitive to sample size [28]. We observed a nonuniform regional distribution of task modulation of GSCORR. A large number of regions, centered in somatosensory, showed GSCORR reduction during the task states, whereas only some regions exhibited task-related increase or remained unchanged (Fig 2B).

We further tested whether the rest-task modulation of GSCORR (i.e., reduced, increased, or unchanged) is task unspecific or task specific. We thereby generated the overlapping maps and calculated the number of task modulations (i.e., reduced, unchanged, or increased) across the seven tasks for each grayordinate (Fig 2C top panel) and the percentages of grayordinates for the number of each modulation (Fig 2C bottom panel). For those regions showing GSCORR reduction, 54.9% regions showed reduction in at least four tasks (4–7 tasks), whereas only 15.5% regions showed GSCORR reduction in 1–3 tasks. For those regions that remained unchanged, 37.8% exhibited similar GSCORR values across at least four tasks (4–7 tasks), whereas 33.4% of these regions remained unchanged in 1–3 tasks. Finally, we refrained from making this calculation in the regions that showed GSCORR increase, as their number was rather small and inconsistent (20.1% in 1–3 tasks and 4.5% in 4–7 tasks) across the various tasks (Fig 2C bottom panel). Together, these findings suggest that the dominant rest-task modulation of GSCORR was task-unspecific decrease, as it mostly occurs across different tasks.

We lastly examined the regional specificity of GSCORR in rest-task modulation. GSCORR reductions were observed mainly in the auditory and somatosensory regions as well as in the regions belonging to the default mode network (DMN). In contrast, the task-related regions exhibited unchanged (plus a small set of increased) modulation, including the primary visual cortex (six of the seven tasks included visual components/stimuli) and the regions belonging to frontoparietal executive control network and ventral attention network (most of the tasks involved some executive and/or attentional function) (Fig 2C top panel).

Taken together, we observed that the GSCORR topography was modulated during different tasks in mostly task-unspecific ways. Such rest-task modulation was mainly manifested in

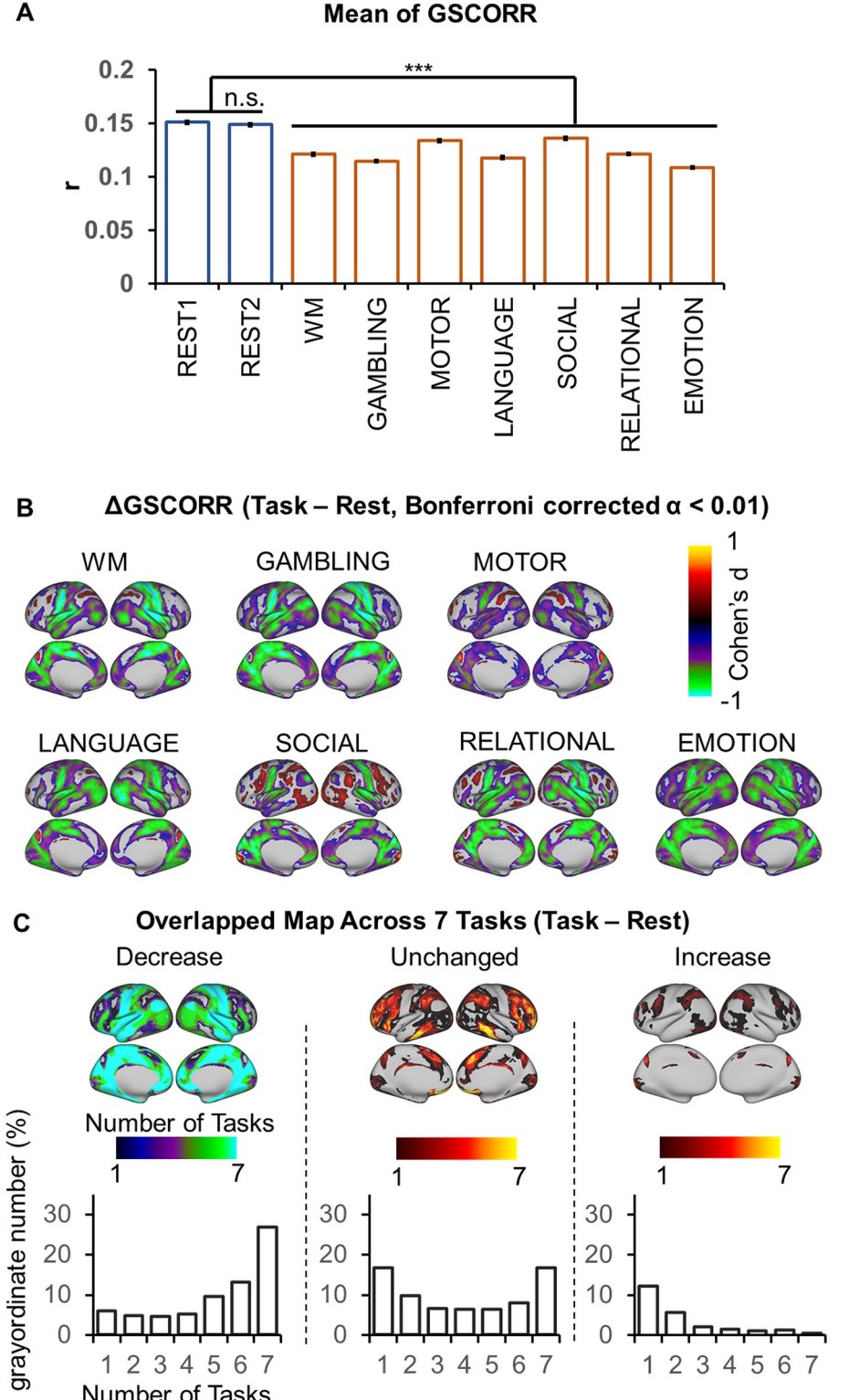

**Fig 2. Rest-task modulation of global signal topography.** (A) Overall GSCORR in day 1 (REST1) and day 2 (REST2) resting state and seven tasks. (B) Grayordinate-based group comparison for GSCORR. The maps were tested with paired *t* tests between task and rest, thresholded at Bonferroni-corrected α < 0.01, and illustrated by Cohen's d to inspect the effect size. (C) Top panel: the overlapping map across the seven tasks illustrated the counted number of each grayordinate showing decrease (left), unchanged (middle), and increase (right) in the tasks when comparing with

rest at the level of Bonferroni-corrected α < 0.01. Bottom panel illustrated the percentage of grayordinates for each counted number in the overlapping maps. ***$p < 0.001$. Data are available at Dryad: https://doi.org/10.5061/dryad. xsj3tx9bw. GSCORR, global signal correlation; n.s., not significant; WM, working memory.

GSCORR reduction in a large set of regions, whereas only some regions exhibited task-related GSCORR increase (and others remained unchanged).

## Instantaneous CAP of GS

Previous studies have shown that selective averaging of fMRI frames exhibiting regional peaks of BOLD activity spatially mimics rs-fMRI networks obtained via static seed-based correlation analysis [29–31]. A recent study has shown that this relationship may also hold true for the GS [6], suggesting that, instead of heightened static correlations between regions, GS topography may be traced to the dynamic CAPs that occur over brief epochs [30]. Therefore, we hypothesized that rest-task modulation of GS topography may be traced to the dynamics of transient coactivation at the peak time points of the GS.

The CAPs of GS were extracted by simply averaging the time points at the top 17% GS (GS-peak), the same threshold as used in a previous study [6] (Fig 3A). The topography of GS-peak resembled that of GSCORR (Fig 3B), with both showing an analogous rest-task modulation pattern (Fig 3C). By inspecting the spatial similarity between GSCORR and GS-peak, we observed that the diagonal value of the similarity matrix reached around 1, suggesting that these two measurements (GSCORR, GS-peak) provide almost identical information about GS spatial pattern. Moreover, our data suggest that rest-task modulation in GS topography is closely related to the dynamic transition of CAP over brief epochs (Fig 3D).

## Decomposing GS-peak into a subset of CAPs

Although we observed that the GS topography reflected an instantaneous coactivation at the peak time points of GS, it is important to understand whether the GS topography is a single united entity or a combination of different coactivation sets. To address this question, we adopted a data-driven approach (i.e., k-means clustering algorithm) that partitioned the whole-brain frames into spatially congruent CAPs (Fig 4A) and assigned each fMRI frame to a cluster label [20,32]. The occurrence rate of these CAPs at the peak time point of GS was measured by the number of CAP occurrences divided by the total number of peak time points (Fig 4A).

As previous studies have demonstrated that a few recurring dominant network states explain the vast majority of rs-fMRI temporal dynamics in both mouse [20] and human [31,32], we here single out the spatial patterns of six and eight CAPs. We determined six CAPs as the optimized number, because these six CAPs yielded previously partitioned large-scale rs-fMRI networks into three pairs of "mirror" motifs and were dissociable across pairs (Fig 4C, see S1 Fig for results of eight CAPs) [20,32]. For example, the CAP1 and CAP4 clearly showed a pair of co-(de)activation networks in somatosensory (CAP1) and frontoparietal network (CAP4). The CAP2 and CAP5 showed a pair of visual (CAP2) and ventral attention network (CAP5). And the CAP3 and CAP6 were a pair of default-mode network (CAP3) and dorsal attention network (CAP6) (Fig 4B).

We next examined the difference of CAP occurrence rates at the time points of the GS-peak between resting state and tasks. During the resting state, the topography of GS-peak was mainly constituted by CAP1 (18%) and CAP2 (22%). During tasks, the contribution of CAP1 increased to 33% (Δrates = 15.6%, $p < 0.001$) on average and the contribution of CAP2 decreased to 8% (Δrates = −13.4%, $p < 0.001$) (Fig 4B and S2 Fig for percentages in each task).

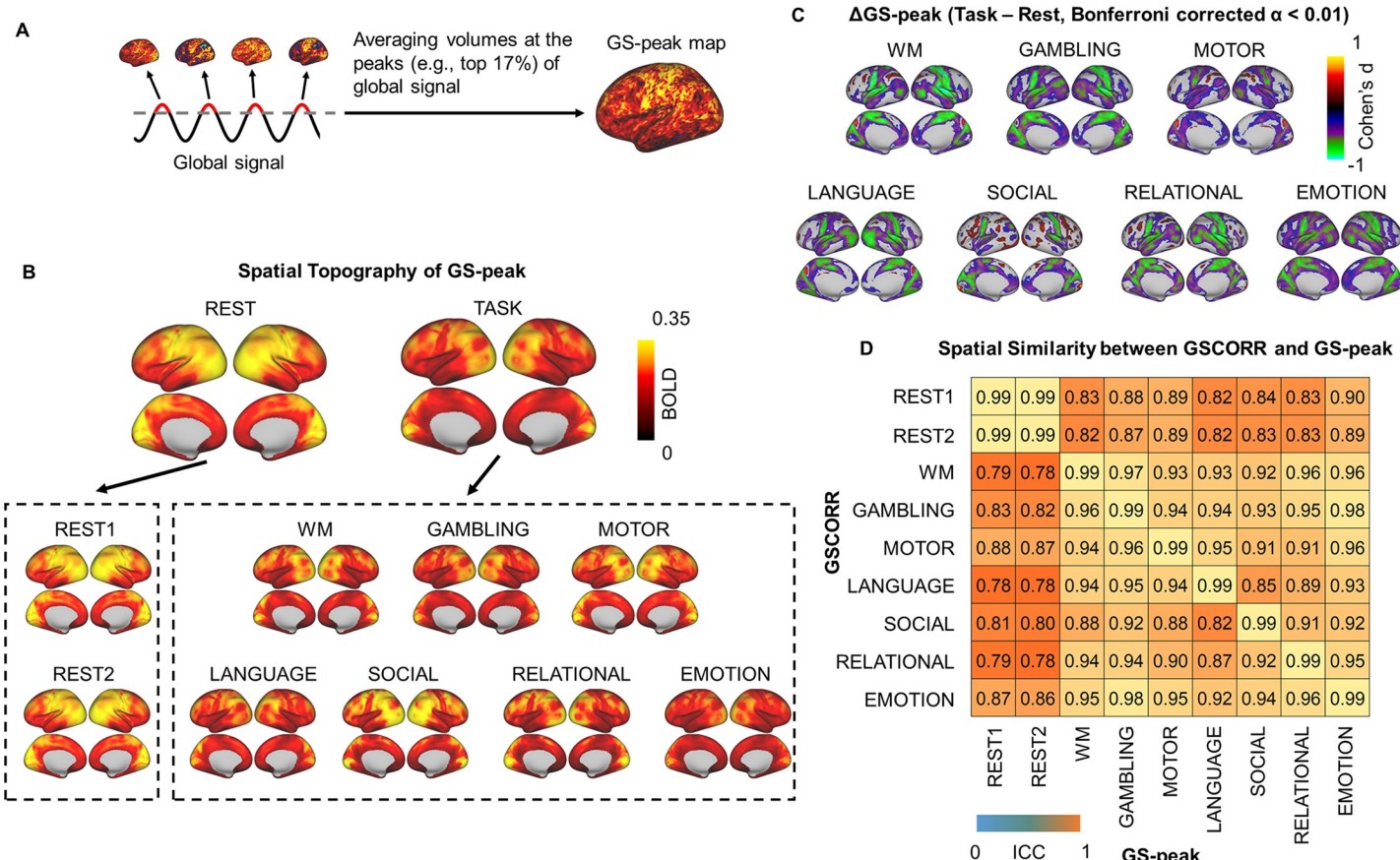

**Fig 3. Instantaneous activation topography at the peak of GS.** (A) An illustration of calculating the coactivation pattern at the peak (top 17%) of the GS ("GS-peak"). (B) Spatial patterns of GS-peak in the 2 days' resting states and the seven tasks. Top panel yielded the averaged spatial patterns for rest and task, respectively. Bottom panel yielded the spatial pattern in each condition. (C) Grayordinate-based group comparisons for GS-peak. The spatial maps were tested with paired *t* tests, thresholded at Bonferroni-corrected α < 0.01, and illustrated by Cohen's d to account for the effect size. (D) ROI-based spatial similarity between GSCORR and GS-peak. Data are available at Dryad: https://doi.org/10.5061/dryad.xsj3tx9bw. BOLD, blood oxygen level–dependent; GS, global signal; GSCORR, GS correlation; ROI, region of interest; WM, working memory.

Finally, we examined whether the rest-task modulation of GSCORR topography is associated with a shift toward any particular CAPs. The spatial correlation between the ΔGSCORR and CAP1 showed significant positive correlations (r = 0.33 on average, *p* < 0.001), and the CAP4, as its opposite, showed significant negative correlation (r = −0.36 on average, *p* < 0.001), whereas the ΔGSCORR and CAP2 showed significant negative correlations (r = −0.66 on average, *p* < 0.001) and its opposite (CAP5) showed significant positive correlations (r = 0.67 on average, *p* < 0.001).

Together, these findings suggested that the GS topography is constituted by a combination of several CAPs that were modulated in the frequency of their occurrence during the tasks.

## Spatial topography of respiration effect

Previous studies showed that the GS in the resting state is closely related to respiration fluctuations [3,16,17]. We therefore investigated the relationship between the GS and respiration time series through their respective spatial topographies and rest-task modulation. If the GS indeed mirrors the fluctuations of respiration, one would expect to observe a comparable spatial topography and rest-task modulation between the respiration and GS.

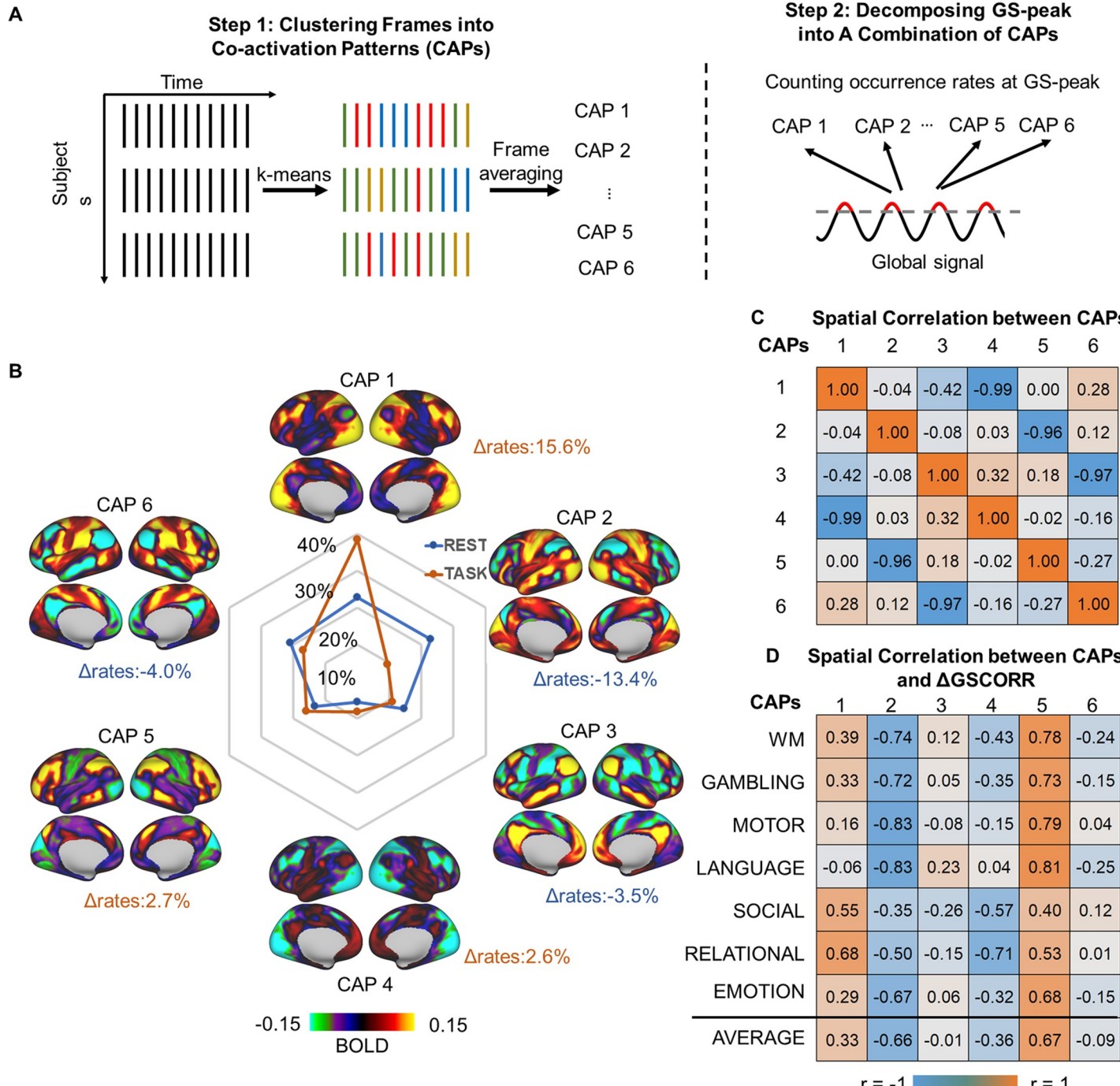

**Fig 4. Transient CAPs at GS-peak.** (A) An illustration of the k-means clustering algorithm that partitions the whole-brain frames into spatially congruent CAPs (step 1), calculating the occurrence rate of each CAP at GS-peak (step 2). (B) Spatial topography of CAPs and their occurrence rates at the time points of GS-peak. Task modulation denoted the difference in the occurrence rate of the CAPs between task and rest (Δrate). (C) Spatial correlation between the CAPs. The six CAPs were composed of three pairs of opposite CAPS (i.e., CAP1 versus CAP4, CAP2 versus CAP5, and CAP3 versus CAP6), as shown by their spatially negative correlations. (D) Spatial correlation between the CAPs and the rest-task difference in GSCORR, i.e., ΔGSCORR. Data are available at Dryad: https://doi.org/10.5061/dryad.xsj3tx9bw. BOLD, blood oxygen level–dependent; CAP, coactivation pattern; GS, global signal; GSCORR, GS correlation; WM, working memory.

We first transformed the respiration time series into respiration volume per time (RVT) (see details in Materials and methods), which has been shown to be closely related with the GS

[3,16,17]. We performed cross-correlations over the time lags from −72 to 72 seconds between the GS and RVT and then grouped into rest and task states. The highest negative correlation was observed around a 12-second lag, which was similar to the previous findings of the negative relationship between GS and RVT [3,33]. However, the GS-RVT correlation showed differences between resting and task states, in which overall weaker correlations were seen in the task states around the time interval ranging from −38 to 18 second (paired $t$ test, Bonferroni-corrected $\alpha < 0.01$) (Fig 5A).

We next calculated the RVT topography by correlating the flipped (reversing the negative into a positive value, to render the RVT topography comparable to GS topography given the negative correlation) RVT signal with the time series in each region with a 12-second lag. The RVT correlation (RVTCORR) showed a weak (RVTCORR ranging from 0 to 0.05, whereas GSCORR ranged from 0 to 0.3, Figs 1B and 5B) but stable (spatial similarity between 2 days' rest: ICC = 0.9913, 95% CI 0.9901–0.9923, Fig 5C) spatial pattern in the resting state. In contrast, the spatial patterns of RVTCORR varied (Fig 5B) and were less similar (ICC = 0.3983, 95% CI 0.3712–0.4264, Fig 5C) across tasks, suggesting an unstable topography of RVTCORR during tasks. The spatial pattern of rest-task modulation revealed significant reductions in somatosensory and visual regions during tasks, and those regions were the ones with higher GSCORR in the resting state (Fig 5D).

Finally, to quantify the contribution of respiration to the GS, we investigated the spatial similarity between GSCORR and RVTCORR within conditions, i.e., resting state and tasks (Fig 5E). The spatial pattern of RVTCORR resembled the pattern of GSCORR in the resting state (ICC = 0.8607, 95% CI 0.8437–0.8760), whereas that was not the case during the tasks (ICC = 0.57, 95% CI 0.5271–0.6109). These results confirmed the major contribution of respiration to the GS in especially the resting state, as observed in previous studies [3,16,17]. In contrast, the contribution of respiration to GS topography was significantly reduced during the tasks, suggesting a dissociation of GS and respiration.

## Noncortical GS components to rest-task modulation—Contributions from basal forebrain

The basal forebrain, reflecting the level of arousal, has been suggested as one of the neural origins of the GS [6,7]. We therefore investigate whether the rest-task modulation of GS topography is driven by this subcortical region. We first replicated the previous findings in both resting states and all tasks (Fig 6, top panel) by showing a negative correlation between the basal forebrain and GS [6,7]. Extending these results, we demonstrated that their correlations did not significantly differ during different tasks when compared with the resting state (see Fig 6 bottom panel). This suggested that the level of arousal is stable across resting and task states, and the rest-task modulation of GS on the cortical level, as described above, may not be driven by the basal forebrain.

## Noncortical GS components to rest-task modulation—Contributions from physiological noise

As the GS may contain various types of noise [3], we checked whether the rest-task modulation of GS topography was affected by head motion, respiration, cardiac, and signals from the ventricle and white matter. We calculated the GSCORR under three procedures of noise regression: (1) without any noise regression, (2) regressors including head motion and signals from the ventricle and white matter as well as their first order derivatives, (3) regressors including those in (2) plus the RVT and heart rate (HR). If the task modulation of GSCORR (i.e., ΔGSCORR) was driven merely by noise, then its spatial pattern, as calculated without noise,

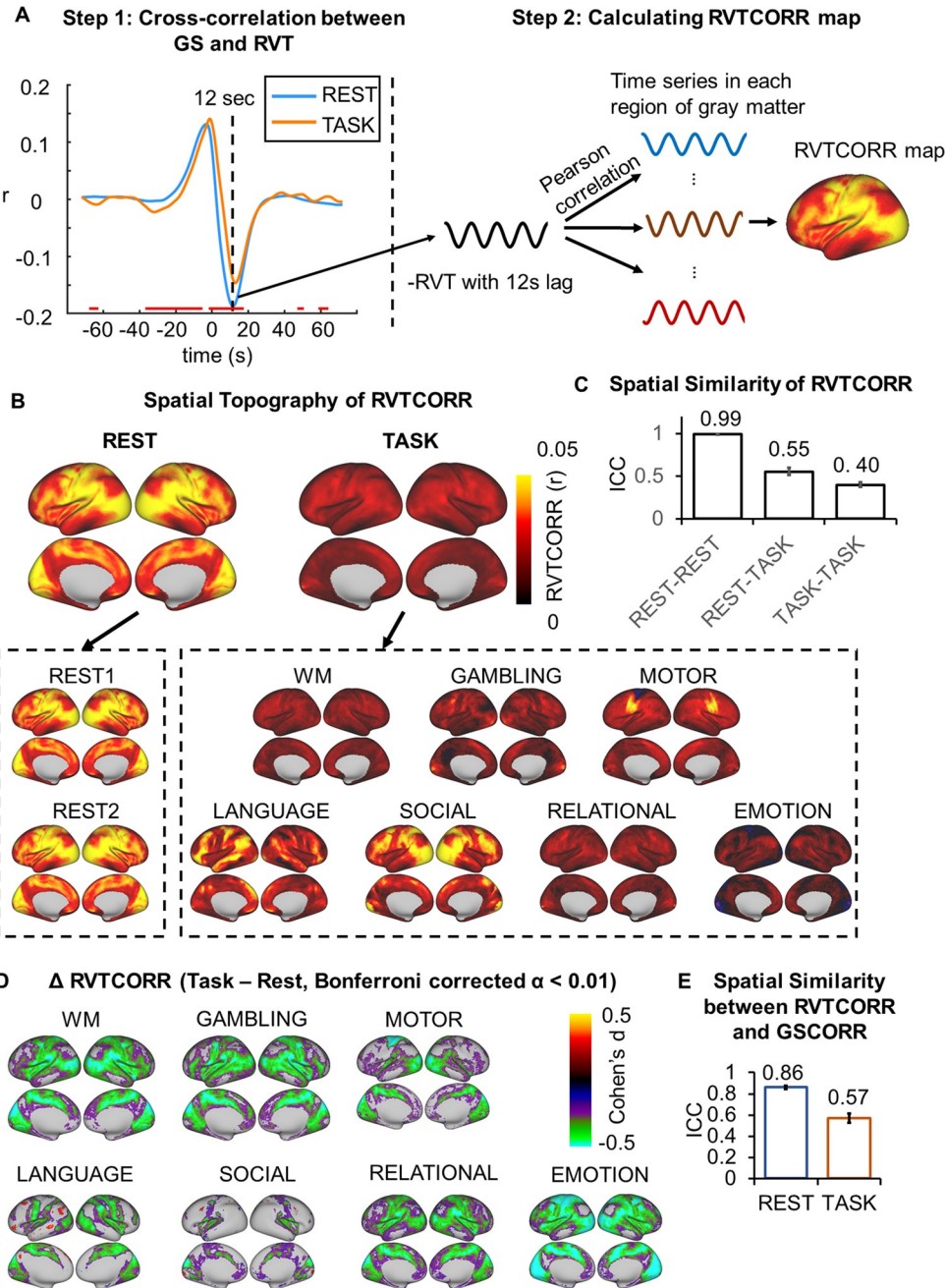

**Fig 5. Spatial topography of the respiration effect.** (A) An illustration of cross-correlation between the GS and RVT, calculating a correlation map between the time series of respiration and time series in each region of the gray matter. The respiration time series was first transformed into the RVT, as the difference between the upper and lower envelope (see Materials and methods for details). Next, the cross-correlation was performed between the GS and RVT to determine the time lag with the strongest correlation, i.e., 12 second (step 1). The RVT was flipped and correlated with the time series in gray matter with 12-second lag (step 2) to obtain the RVTCORR map. The time interval showing significant rest-task differences of the GS-RVT correlations was indicated by red line, thresholded at Bonferroni-corrected α < 0.01. (B) Spatial patterns of RVTCORR in 2 days' resting states and seven tasks. Top panel corresponded to the averaged spatial patterns for resting and task states, respectively. Bottom panel corresponds to the spatial pattern in each condition. (C) The spatial similarity (ICC ± 95% CI) of RVTCORR between the 2 days' resting states, between the resting and task states, and between the tasks. (D) Rest-task modulation of RVTCORR. (E) Spatial similarity (ICC ± 95% CI) between GSCORR and RVTCORR for resting and task states, respectively. Data are available at Dryad: https://doi.org/10.5061/dryad.xsj3tx9bw. GS, global signal; GSCORR, GS correlation; ICC, intraclass correlation coefficient; RVT, respiration volume per time; RVTCORR, RVT correlation; WM, working memory.

# GSCORR at Basal Forebrain

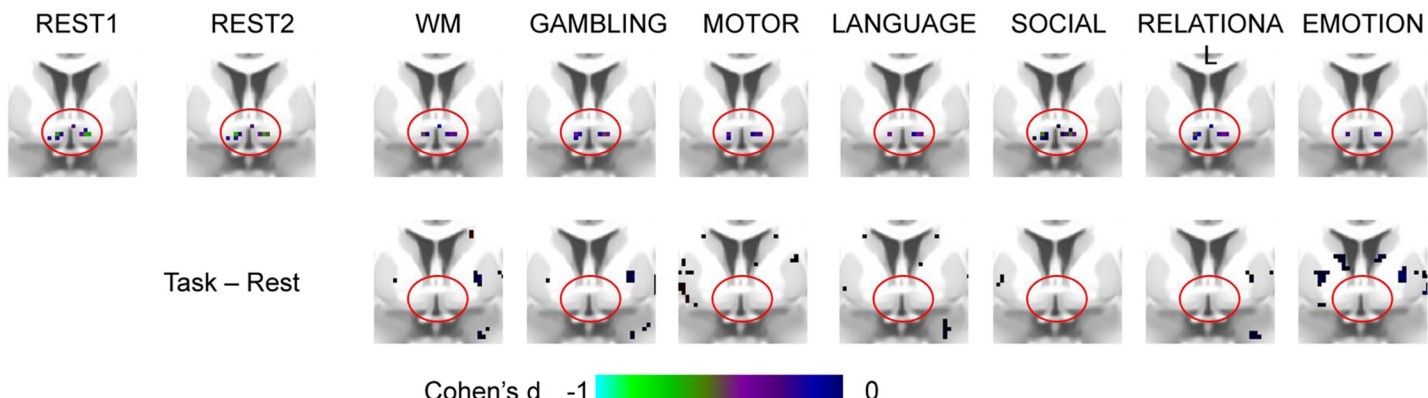

**Fig 6. GSCORR at basal forebrain in rest and tasks.** The maps are thresholded at Bonferroni-corrected α < 0.01. Top panel, group GSCORR comparisons with baseline (H0: r = 0). The negative correlations are consistently seen across the 2 days' resting states and all task states. Bottom panel, group GSCORR comparisons of task versus rest show an absence of any significant voxel in the basal forebrain. Data are available at Dryad: https://doi.org/10.5061/dryad.xsj3tx9bw. GSCORR, global signal correlation; WM, working memory.

should show low degrees of spatial similarity with the one after noise regression. We found a high spatial similarity of the ΔGSCORR before and after nuisance noise regression (ICC > 0.95 in all cases) (Fig 7). In addition, we observed that the inclusion of noise regression (i.e., head motion, white matter, and cerebrospinal fluid [CSF]) increased the effect size of ΔGSCORR in all seven tasks (increased percentage in Cohen's d ranging from 22% to 39%) but was not further improved when adding respiration and cardiac recording.

Together, these results suggested that GS topography during rest-task modulation cannot be explained by nonneuronal components in the GS. Instead, regressing out nonneuronal components increased the signal-to-noise ratio (SNR) and strengthened the presumed neuronal component of GSCORR in the measured signal.

## Reproducibility of the GS topography and its rest-task modulation

To examine the reliability of our findings, we calculated the GSCORR, GS-peak, and the occurrence rate of CAPs in an independent 7T dataset from HCP. This dataset included four sessions of resting state, four sessions of movie watching, and six sessions of retinotopy stimuli perception. Similar to the findings in the 3T dataset, we observed higher GSCORR and GS-peak in somatosensory and visual cortex during the resting state (Fig 8A). Moreover, rest-task modulation also showed (as seen in Figs 2B and 3C for the 3T dataset) a reduction of GSCORR and GS-peak in the somatosensory cortex (Fig 8B). Finally, the CAPs showed a comparable, albeit smaller, effect of rest-task modulation, with increased reoccurrence rate in CAP1 and decreased reoccurrence rate in CAP2 (Fig 8C).

## Discussion

In this study, we demonstrated the informative nature of the GS through various lines of data on rest-task modulation of its topography. We first showed that the spatial pattern of GS was modulated during tasks with reduced contributions of somatosensory regions to GS topography. This rest-task modulation was captured by the dynamics of GS topography in terms of transient CAPs that occur over brief epochs at the peak periods of the GS. By comparing the spatial similarity between the GSCORR and the ones observed in relation to respiration, we

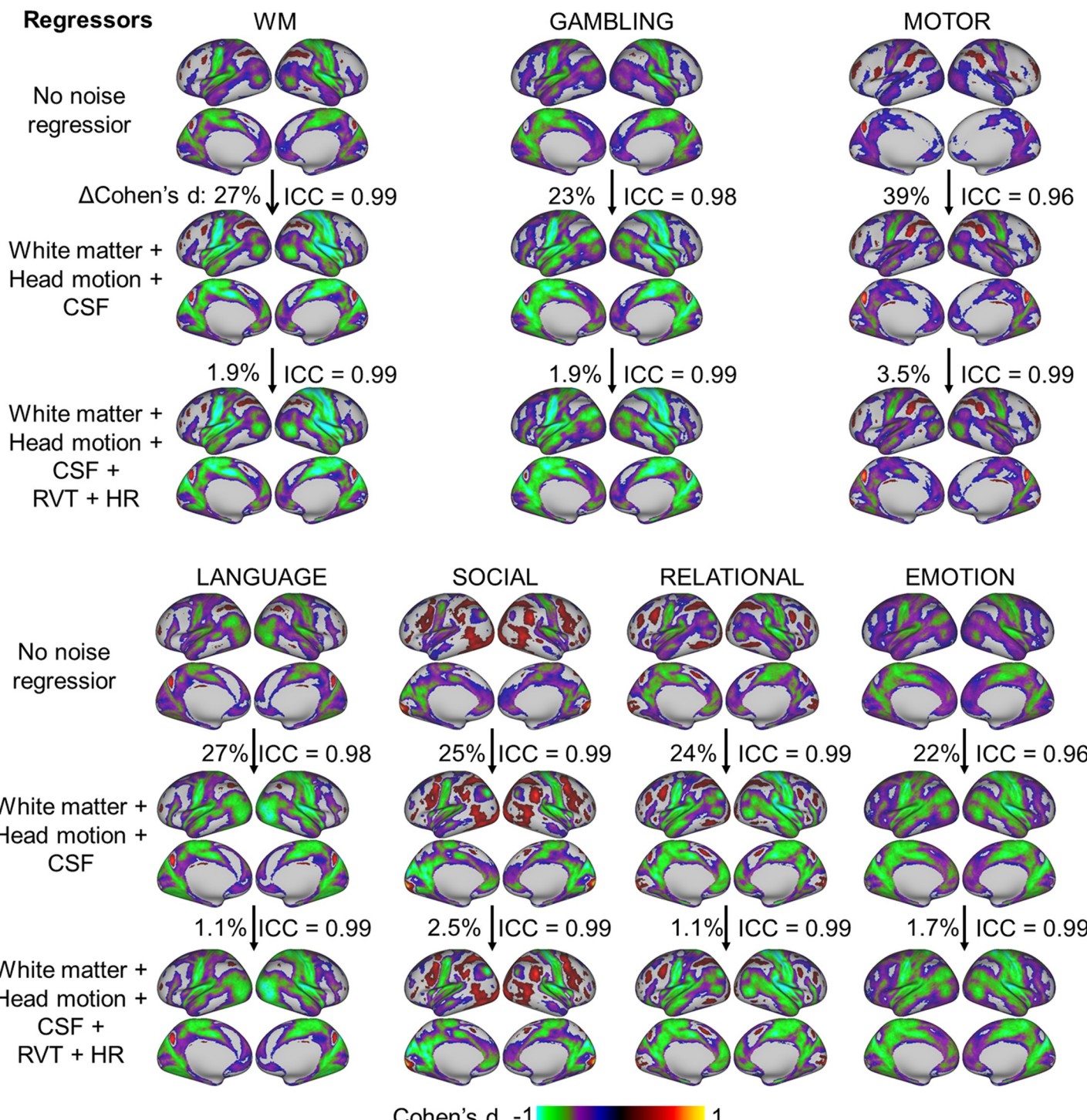

**Fig 7. Rest-task modulation of GSCORR topography under different procedures of noise regression.** The maps were thresholded at Bonferroni-corrected α < 0.01. The spatial similarity of rest-task modulation, i.e., ΔGSCORR, was measured based on the Cohen's d GSCORR topography. The percentage of increased effect size was calculated as the increased overall |d| value (relative to the d value from the previous step) divided by the |d| value from the previous step. Data are available at Dryad: https://doi.org/10.5061/dryad.xsj3tx9bw. CSF, cerebrospinal fluid; GSCORR, global signal correlation; HR, heart rate; ICC, intraclass correlation coefficient; RVT, respiration volume per time; WM, working memory.

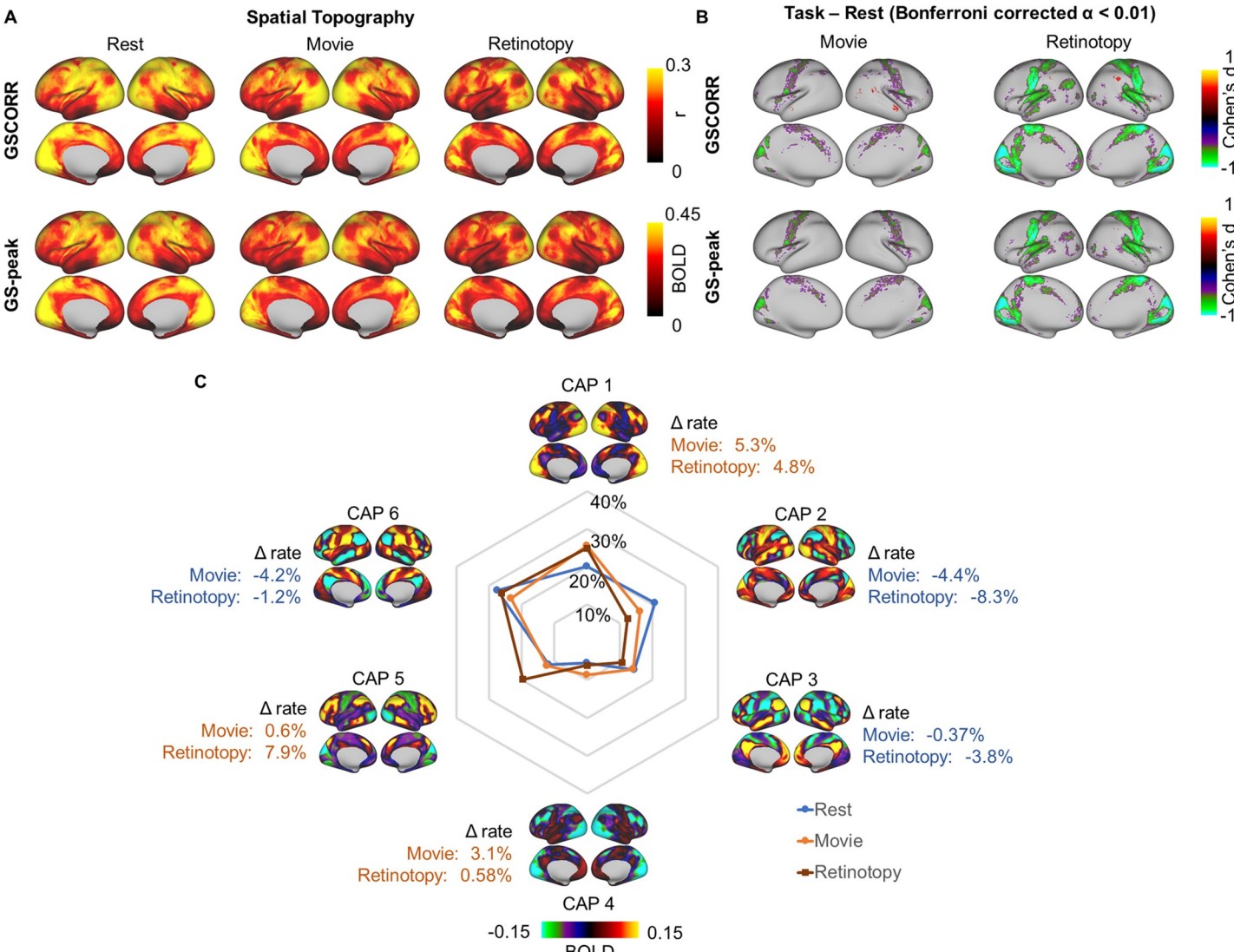

**Fig 8. Reproducibility of the GS topography and its task modulation in HCP 7T dataset.** (A) Spatial topography measured by GSCORR and GS-peak in resting state and two additional tasks (e.g., movie watching and retinotopy stimuli watching). (B) Grayordinate-based group comparison for GS topography. The maps were tested with paired *t* test between each task and rest. The maps were thresholded at Bonferroni-corrected α < 0.01 and illustrated by Cohen's d. (C) Occurrence rates of CAPs and the difference between tasks and resting state. Data are available at Dryad: https://doi.org/10.5061/dryad.xsj3tx9bw. BOLD, blood oxygen level–dependent; CAP, coactivation pattern; GS, global signal; GSCORR, GS correlation; HCP, Human Connectome Project; T, tesla; WM, working memory.

observed that the topography of respiration mimicked the topography of the GS mainly for the resting state but significantly less so in the tasks; we therefore supposed that the GSCORR could not be equated with respiration. Finally, we ruled out the potential contributions from subcortical areas or other sources of physiological noise on the rest-task modulation of GS topography.

## GS topography—Spatial distribution and rest-task modulation

We found that the GS exhibits a nonuniform topographical distribution across brain regions, characterized by higher levels of GSCORR in primary sensory regions like auditory and visual cortices. In contrast, we observed lower levels of GSCORR in higher-order cortical areas like

the prefrontal cortex. This replicated recent findings in both monkeys [7] and humans [3,6,11–13]. In our study, we extended these resting-state observations to the changes of GS topography in different tasks, i.e., rest-task modulation. If the GS is subject to artifactual noise —i.e., nothing but noise—one would expect no rest-task modulation of GS topography, as the noise level may remain the same during both resting and task states. If, in contrast, the GS reflects neuronally relevant information, one would expect modulation effects of its topography during task states.

Our results support the latter. Specifically, we demonstrated that the GS topography was modulated during different tasks in largely task-unspecific ways—that is, in more or less the same manner during the seven different tasks. Moreover, we observed predominant task-unspecific decreases of GSCORR in task-irrelevant regions (e.g., somatosensory cortex) during the tasks. In contrast, the task-related regions like the primary visual cortex (as almost all the tasks included visual stimuli) exhibited unchanged GSCORR or, in a few regions, an increase in GSCORR.

Together, we conclude that the GS does indeed provide relevant information about the brain's neuronal activity in different states, i.e., rest and task. Therefore, we consider such rest-task modulation as first novel line of support for the informative nature of GS and its GS topography.

## Dynamics of GS topography—Transient CAPs

The GS topography is commonly measured by GSCORR [1,3,10], linear regression beta weights between the GS and time series in each voxel [11,12], or whole-brain functional connectivity [14,15]. Recently, an analogous spatial pattern has also been observed in the instantaneous CAP at the peak time points of the GS during the resting state [6]. Building on that, another study demonstrated that some CAPs are phase-locked to the stimulus onset during task states [32]. Together, these findings suggest that the GS topography is not static but rather dynamic, consisting in a set of transient CAPs over brief epochs, which can be modulated during task states.

In order to support the hypothesis of the dynamic nature of GS topography, we conducted the same kind of analyses of CAPs during the peak of the GS for both resting and task states. We first found a high spatial similarity between the traditionally calculated static GS topography and the CAP at the peaks of GS. Next, using a data-driven clustering approach, we demonstrated that the averaged CAP pattern during the GS-peak could be decomposed into a set of finer-grained CAPs (see [20] for showing analogous findings in animals). Most importantly, we showed that these dynamic CAPs are modulated during task states, in terms of a change in the frequency rates. For instance, the occurrence rates of CAP1 increased while that of the CAP2 decreased. Summing up, our observation of the dynamic nature of GS topography in terms of transient CAPs further support the assumption of the informative nature of GS.

Although we demonstrated that the rest-task modulation of GS was associated with alterations of the prevalence of CAPs, an alternative interpretation (or a potential confound) of the results may be considered. More specifically, the "resting state" has (trivially) no task and no temporal structure, whereas each of the task states trivially does. This task structure may impose temporal structure on both the GS (due to task-linked breathing and HR changes) and task-linked BOLD responses (e.g., from visual or other stimuli associated with the task, responses such as button presses, which are necessarily entrained to the task); both GS and task-linked BOLD would have significant components along the vector defined by the response time course, making them hard—if not impossible—to separate. Simply comparing estimates of GS during task versus rest would be confounded by this difference in temporal

structure of rest and task states. Because within the BOLD signal both entrainment to the temporal structure of the task and task-linked neural activity are unavoidably correlated in time, it remains methodologically rather challenging to decompose these two components in a meaningful way.

## Subcortical and physiological substrates underlying GS topography

The GS has been shown to be closely related to physiological signals. While controlling the head motion and scanning noise, the GS has been shown to exhibit high correlation with the respiration rate [3,16,17,33]. We replicated the close relationship of respiration of GS by demonstrating their spatial similarity during resting state. This suggests that the GS is indeed coupled to respiration. Remarkably, we observed that the coupling of GS to respiration in terms of their spatial similarity pattern was significantly decreased during all task states. However, caveats still need to be taken for interpreting this effect. For example, breathing (and HR) may be strongly entrained to task timing when the participants are engaged in a task. The GS due to breathing (and HR) could thus have components that correlate or anticorrelate with the neurally driven task-related response in a relevant brain area. Therefore, the interplay among the GS, task-evoked brain activities, and physiological signals may be an important question that warrants more investigations.

Another source contributing to the GS, as observed by recent studies, is the basal forebrain, as supported by empirical data in both human [6] and macaque [7]. This region is functionally involved with arousal or vigilance regulation [34–37]. We did not find statistically significant difference of the rest-task modulation in the basal forebrain contributions to GS. However, caveats need to be taken for interpreting the null effect. That is, the absence of a rest-task difference may not be evidence for a lack of involvement of the basal forebrain. First, the basal forebrain is a very small structure with poor SNR [6], which may contribute to a type II error. Second, our analytical approach may be not sensitive enough to detect the basal forebrain's contribution under the broad differences between resting and task states.

Nevertheless, from another angle, our results support the findings by Liu and colleagues [6] in terms of the strong negative correlation between the basal forebrain and GS in general. We found that the negative correlation is robust across both the resting and task states. The seemingly invariant negative correlation may indicate that the arousal level is similar across the resting and task states, and rest-task modulation most likely occurred on the cortical level that is associated with cognitive processing.

Finally, the GS may also, in part, include widespread nonneuronal noise [3,21]. We therefore checked the influence of noise regression on rest-task modulation of GS. Our data show highly correlating and thus similar topographical distribution of rest-task GS modulation both with and without physiological noise regression. If the GS topography were reflecting only the spatial pattern of nonneuronal noise, one would have expected the opposite—namely, low or no correlation in GS topography between analyses with and without physiological noise regression.

More importantly, we also demonstrated that regressing the noise can significantly improve the SNR in detecting rest-task modulation of GS topography; that further supports the need for noise regression before calculating GS topography. Ideally, future investigations are warranted to combine rest-task modulation in fMRI with electrophysiological investigation to further corroborate its assumed neuronal basis.

## Clinical relevance—GS topography in psychiatric disorders

The rest-task modulation of GS topography may provide a novel framework for interpreting symptoms in psychiatric disorders. The GS topography during resting state has been

investigated in different psychiatric disorders like schizophrenia, bipolar disorders, and autism spectrum disorders [10,11,14,15,38]. For example, the data showed altered GS topography with abnormal GS increase in association regions in schizophrenia [11] as well as enhanced GS contribution from motor regions in bipolar manic participants and from hippocampus/parahippocampus in bipolar depression [10].

Albeit tentatively, GS increases and decreases in these regions may lead to corresponding increases or decreases of the functions that are associated with the respective regions. For instance, schizophrenia can be characterized by abnormal cognitive processing as in delusions and formal thought disorders that are known to be related to abnormalities in that association cortex that also shows GS increase [11]. Or GS increase in motor cortex of manic patients may be related to increased motor function, which symptomatically is manifest in psychomotor agitation. Finally, depressed patients exhibit typically rumination with increased recollection of autobiographical memories [10], which may be related to their increased GS in hippocampus/parahippocampus.

Since these GS abnormalities were all observed during the resting state, one may assume that these symptoms may neuronally be traced to virtual task states [39] in the respective regions during resting state. If so, the psychiatric symptoms may mirror the "normal" cognitive function of these regions in an "abnormal" neuronal context—that is, task-like state during the resting state. One would then expect that, unlike in healthy individuals (as in the present study), psychiatric patients may no longer show rest-task modulation of GS topography in specifically these regions nor the task-related changes in the occurrence rates of their dynamic CAP. Psychiatric symptoms may ultimately be traced to the confusion of resting and task states as modulated by abnormalities in GS topography, thus entailing what recently has been described as "spatiotemporal psychopathology" [40–43].

## Materials and methods

### Ethics statement

This paper utilized data collected for the HCP. The scanning protocol, participant recruitment procedures, and informed written consent forms, including consent to share deidentified data, were approved by the Washington University institutional review board [44].

### Data collection

We utilized the dataset from the HCP. The 3T dataset was served for main analyses, and 7T served for replication [44,45]. All individuals from the HCP 1,200-participant data release (March 1, 2017) having completed fMRI sessions were included (837 participants in 3T and 145 participants in 7T, respectively). The age of the participants in the project ranged from 22 to 35 years, and 54% were female.

### Procedures of resting state and tasks in 3T

Data were collected over 2 days. On each day, the data collection included 28 minutes of rest (eyes open with fixation) fMRI data across two runs and 30 minutes of task fMRI. Each of the seven tasks was completed over two consecutive fMRI runs [46]. These tasks included seven distinct domains: emotion, reward learning, language, motor, relational reasoning, social cognition, and working memory (WM). Briefly, the emotion task involved matching fearful or angry faces to a target face. The reward-learning task involved a gambling task with monetary rewards and losses. The language task involved auditory stimuli consisting of narrative stories and math problems, along with questions to be answered regarding the prior auditory stimuli.

The motor task involved movement of the hands, tongue, and feet. The relational reasoning task involved higher-order cognitive reasoning regarding relations among features of presented shape stimuli. The social cognition task used short video clips of moving shapes that interacted in some way or moved randomly, with participants making decisions about whether the shapes had social interactions. The WM task consisted of a visual N-back task, in which participants indicate a match of the current image to either a constant target image or two images previous. The details of tasks can be found elsewhere [46].The whole-brain echo-planar imaging acquisitions were acquired with a 32-channel head coil at a resolution of 2-mm isotropic and 0.72-second TR [47].

## Procedures of resting state and tasks in 7T

Data collection in 7T contained a total of 14 fMRI runs, including 64 minutes of rest, 64 minutes of movie watching, and 30 minutes of retinotopy [47,48]. In each movie-watching run, participants had to watch a movie of approximately 15 minutes consisting of several short clips separated by 20-second rest periods. Different clips were used in different runs (details are available at HCP S1200 Release Reference). For retinotopy task, stimuli were constructed by creating slowly moving apertures and placing a dynamic colorful texture within the apertures [49]. The data were collected at a resolution of 1.6-mm isotropic and 1-second TR (multiband acceleration 5, in-plane acceleration 2, 85 slices) [47].

## Preprocessing

Preprocessing was carried out using Workbench [50] and custom code in MATLAB 2017b (MathWorks). For 3T data, to match the steps of preprocessing across rest and task, the version of HCP minimal preprocessing pipeline before FIX denoising was used (i.e., including procedures of registration to MNI space, alignment for motion, fieldmap correction, and MSMAll group registration [47]). Additional noise regression was applied by in-house code (see details below). For 7T data, as the minimal preprocessed data in both rest and task had performed FIX denoising [47], no additional noise regression was further applied.

For 3T data, the linear trend for each run was removed, and the nuisance time series (ventricle, white matter, motions along with their first order derivatives) were regressed by using linear regression [3,51]. The nuisance time series (ventricle and white matter signals) were extracted from volume-based minimal processing. No low-pass temporal filter was applied, given the possibility that frequency specificity might differ between resting and task state [51].

## Preprocessing of physiological recordings

The cardiac and respiration recordings in the 3T dataset were analyzed. The preprocessing followed the previous studies [3,16,17,33]. More specifically, the respiratory signal was detrended linearly. The outliers were defined as the time points that deviated more than (approximately) 7 median absolute deviations (MADs) from the moving median value within a time window of 30 seconds. Subsequently, the respiratory signal was low-pass filtered at 5 Hz with a second-order Butterworth filter. The recording was then transformed into the time series of RVT, calculated as the difference between upper and lower envelope [16,17]. Outliers were replaced using linear interpolation. Finally, the RVT time series was resampled to the rate of fMRI recording (TR = 0.72 seconds).

The cardiac signal was initially band-pass filtered with a second-order Butterworth filter between 0.3 and 10 Hz. The peak was detected with minimum peak distance larger than 0.3 seconds. The cardiac signal was then computed in beats per minute (bpm) by multiplying the inverse of the time differences between pairs of adjacent peaks with 60. Outliers were corrected

using the moving median method described above and replaced using linear interpolation. Finally, the HR time series was resampled to the rate of fMRI (TR = 0.72 seconds).

### GSCORR

The GS was calculated for each participant by averaging the standardized (z-score) fMRI signals across grayordinate. The GS topography was calculated by Pearson correlation (i.e., GSCORR) between the GS and the time series in each grayordinate [1,3]. The correlation r values were then transformed through Fisher z transformation for statistical analyses [51,52].

### GS CAP (GS-peak)

The method for calculating the GS-peak topography was adopted from previous studies [6,29]. Time points at the top 17% of GS were selected, and the fMRI volumes at these time points were averaged to generate the GS CAP (GS-peak).

### Whole-brain CAP analysis (CAPs)

To investigate whether the GS coactivation topography was a combination of several specific recurring spatial patterns, we adopted an unsupervised machine-learning approach using k-means clustering algorithm [30,32]. This procedure classified a set of fMRI volumes into different categories (e.g., patterns) into k clusters based on their spatial similarity and thus produced a set of CAPs or brain states. In this way, the original fMRI (3D + time) data were transformed into a 1D time series of discrete CAP labels.

The analysis above was performed on the concatenated data of the 3-T dataset, including resting state and all tasks from all 837 participants. To increase spatial SNR in local brain regions and make the computation achievable in this huge dataset, we performed the k-means clustering at the ROI level [31], using a standard brain parcellation with 1,000 ROIs [27]. After clustering, the fMRI frames assigned to the same cluster were simply averaged, resulting in k maps defined as CAPs. The CAP labels in the 7-T dataset were assigned on its maximal similarity to the predefined CAPs from the 3-T dataset, to make the results comparable and generalizable across datasets [32].

Previous studies demonstrated a few recurring dominant network states explain the vast majority of rs-fMRI temporal dynamics in both mouse [20] and human [31,32]; we here inspected the spatial patterns under six and eight CAPs and found that the six clusters led to the maximal number of identifiable spatial patterns, and these patterns were conserved when the clusters increased to eight (see S1 Fig).

### Respiration topography (RVTCORR)

Cross-correlation over lags (−72 to 72 seconds) was first performed between GS and RVT. The cross-correlation demonstrates the GS and RVT showed strongest negative correlation with an approximately 12-second lag. To make the topography of GSCORR and RVTCORR comparable, the RVTCORR topography was calculated by Pearson correlation between the flipped RVT with 12-second lag and the time series in each grayordinate. The correlation r values were then transformed through Fisher z transformation for statistical analyses [51,52].

### Analysis of noise effect on GSCORR

To examine whether the GS topography differed across states because of the difference of nuisance signal contribution [3], we additionally calculated the GS topography with different procedures of noise regression: (1) without any noise regression; (2) regressors including head

motion and signals from the ventricle and white matter as well as their first order derivatives; and (3) regressors including those in (2) plus the RVT and HR. The grayordinate-based correlation for the task-rest difference was calculated to check the influence of nuisance signal on rest-task difference [53].

## Analysis of spatial similarity

To avoid the correlation induced by spatial adjacency in grayordinate-based correlation [54], the analyses of absolute spatial similarities for GSCORR, GS-peak, and RVTCORR were performed using ICC [23], across the aforementioned 1,000 ROIs at the group level, with spatial normalization (z-score). In reporting these findings, the ICC was categorized into five common intervals [26]: $0 < ICC \leq 0.2$ (slight); $0.2 < ICC \leq 0.4$ (fair); $0.4 < ICC \leq 0.6$ (moderate); $0.6 < ICC \leq 0.8$ (substantial); and $0.8 < ICC \leq 1.0$ (almost perfect). The significant difference between two ICCs was checked by Fisher z test [55].

## Statistics

All statistical inferences for topography were based on two-tailed paired $t$ tests. In the current study, we mainly report results with effect size using Cohen's d [56] instead of the t value, as Cohen's d is independent of the sample size. The threshold for the differences in brain maps was set at Bonferroni-corrected $\alpha < 0.01$ across the whole brain.

## Supporting information

**S1 Fig. Transient CAPs with eight clusters.** (A) Spatial topography of CAPs and their occurrence rates at the time points of GS-peak. Task modulation denoted the difference in the occurrence rate of the CAPs between task and resting state (Δrate). (B) Spatial correlation between the CAPs. The eight CAPs were composed of four pairs of opposite CAPs (i.e., CAP1 versus CAP5, CAP2 versus CAP6, CAP3 versus CAP7, and CAP4 versus CAP8), as denoted by their negative correlations. Data are available at Dryad: https://doi.org/10.5061/dryad.xsj3tx9bw. CAP, coactivation pattern; GS, global signal.
(TIF)

**S2 Fig. Transient CAPs across 2 days' rests and seven tasks.** Top panel yielded the CAPs in Fig 3. Bottom panel yielded the occurrence rate across resting state and seven tasks. Data are available at Dryad: https://doi.org/10.5061/dryad.xsj3tx9bw. CAP, coactivation pattern.
(TIF)

## Author Contributions

**Conceptualization:** Jianfeng Zhang.

**Data curation:** Jianfeng Zhang.

**Formal analysis:** Jianfeng Zhang.

**Methodology:** Jianfeng Zhang, Zirui Huang, Shankar Tumati.

**Resources:** Georg Northoff.

**Supervision:** Georg Northoff.

**Validation:** Jianfeng Zhang.

**Visualization:** Jianfeng Zhang.

**Writing – original draft:** Jianfeng Zhang, Zirui Huang, Shankar Tumati, Georg Northoff.

**Writing – review & editing:** Jianfeng Zhang, Zirui Huang, Georg Northoff.

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
