## [Editor Report · Decision Letter 0]

9 Oct 2019

Dear Dr Zhang, 

Thank you for submitting your manuscript entitled "Intrinsic Architecture of Global Signal Topography and Its Modulation by Tasks" for consideration as a Research Article by PLOS Biology.

Your manuscript has now been evaluated by the PLOS Biology editorial staff, as well as by an academic editor with relevant expertise, and I'm writing to let you know that we would like to send your submission out for external peer review.

Please re-submit your manuscript within two working days, i.e. by Oct 11 2019 11:59PM.

Kind regards,

Roli Roberts

Senior Editor

PLOS Biology

---

## [Decision Letter · Decision Letter 1]

25 Nov 2019

Dear Dr Zhang,

Thank you very much for submitting your manuscript "Intrinsic Architecture of Global Signal Topography and Its Modulation by Tasks" for consideration as a Research Article at PLOS Biology. Your manuscript has been evaluated by the PLOS Biology editors, an Academic Editor with relevant expertise, and by three independent reviewers.

You'll see that two of the reviewers are positive about your study, but reviewer #1 remains to be convinced that your paper represents a significant enough advance over the literature, and has some questions regarding the support for your central claims. Reviewer #3 also has some statistical concerns.

In light of the reviews (below), we will not be able to accept the current version of the manuscript, but we would welcome resubmission of a much-revised version that takes into account the reviewers' comments. We cannot make any decision about publication until we have seen the revised manuscript and your response to the reviewers' comments. Your revised manuscript is also likely to be sent for further evaluation by the reviewers.

Your revisions should address the specific points made by each reviewer. Please submit a file detailing your responses to the editorial requests and a point-by-point response to all of the reviewers' comments that indicates the changes you have made to the manuscript. In addition to a clean copy of the manuscript, please upload a 'track-changes' version of your manuscript that specifies the edits made. This should be uploaded as a "Related" file type. You should also cite any additional relevant literature that has been published since the original submission and mention any additional citations in your response. 

Before you revise your manuscript, please review the following PLOS policy and formatting requirements checklist PDF: http://journals.plos.org/plosbiology/s/file?id=9411/plos-biology-formatting-checklist.pdf. It is helpful if you format your revision according to our requirements - should your paper subsequently be accepted, this will save time at the acceptance stage.

Please note that as a condition of publication PLOS' data policy (http://journals.plos.org/plosbiology/s/data-availability) requires that you make available all data used to draw the conclusions arrived at in your manuscript. If you have not already done so, you must include any data used in your manuscript either in appropriate repositories, within the body of the manuscript, or as supporting information (N.B. this includes any numerical values that were used to generate graphs, histograms etc.). For an example see here: http://www.plosbiology.org/article/info%3Adoi%2F10.1371%2Fjournal.pbio.1001908#s5.

For manuscripts submitted on or after 1st July 2019, we require the original, uncropped and minimally adjusted images supporting all blot and gel results reported in an article's figures or Supporting Information files. We will require these files before a manuscript can be accepted so please prepare them now, if you have not already uploaded them. Please carefully read our guidelines for how to prepare and upload this data: https://journals.plos.org/plosbiology/s/figures#loc-blot-and-gel-reporting-requirements.

Upon resubmission, the editors will assess your revision and if the editors and Academic Editor feel that the revised manuscript remains appropriate for the journal, we will send the manuscript for re-review. We aim to consult the same Academic Editor and reviewers for revised manuscripts but may consult others if needed.

We expect to receive your revised manuscript within two months. Please email us (plosbiology@plos.org) to discuss this if you have any questions or concerns, or would like to request an extension. At this stage, your manuscript remains formally under active consideration at our journal; please notify us by email if you do not wish to submit a revision and instead wish to pursue publication elsewhere, so that we may end consideration of the manuscript at PLOS Biology.

When you are ready to submit a revised version of your manuscript, please go to https://www.editorialmanager.com/pbiology/ and log in as an Author. Click the link labelled 'Submissions Needing Revision' where you will find your submission record. 

Sincerely,

Roli Roberts

Senior Editor

PLOS Biology

REVIEWERS' COMMENTS:

Reviewer #1:

The authors performed global signal correlation (GSCORR) analysis, and found the pattern is stable across resting-state and tasks. They also found tasks can modulate the GSCORR pattern, and claimed this is reflecting phase coherence rather than amplitude variation.

Major concerns:

1. It has no doubt that GSCORR has a stable pattern, as summarized very well in Power et al.’s 2017 paper (Sources and implications of whole-brain fMRI signals in humans). Power’s paper has a clear demonstration of GSCORR pattern and interpretation of sources of GSCORR, but the viewpoints were not thoroughly discussed (although cited) in the current manuscript. So as this Liu et al.’s 2017 paper: The global signal in fMRI: Nuisance or Information?

2. The authors’ main point “the GS topography (GSCORR) exhibited a stable spatial pattern across rest and task, supporting the term as intrinsic architecture” is not innovative. Changing the seed time series from global signal to other seeds (e.g., posterior cingulate cortex for mapping default mode network, or V1 to mapping visual network), will have similar results: stable pattern across tasks but can be modulated by tasks. Even more radically, instead of performing any functional connectivity analysis (correlation analysis), simply averaging all the volumes can have similar effect. However, I don’t think these kind results solved any “fundamental question” (e.g., GS topography is whether it is an intrinsic architecture).

3. The authors’ conclusion, “phase coherence, rather than amplitude variation, serves as the neural basis of GS architecture”, can not be validated through their analyses. To make such a conclusion, the authors compared global signal Kuramoto change and standard deviation change. However, the calculation of global signal Kuramoto is very similar to global signal correlation by definition, the authors can test these two on simulated time series. In standard deviation calculation, no global signal is involved, thus it’s not surprising that this measure will have results more distant to global signal correlation.

“The contribution to global signal from each grayordinate can depend on either the level of amplitude or the level of phase coherence”. The authors directly used the amplitude (standard deviation) of the signal of a certain grayordinate. However, the more important question is how much the global signal was present in the signal of this grayordinate (i.e., the fraction of the amplitude of GS component present in the signal of this grayordinate).

Minor concerns:

1. Multiple comparison correction. For voxel/vertex-wise comparison, the authors “used the threshold at |Cohen’s d| > 0.2 for further analyses (the threshold for reaching small effect size, corresponding to |t-values| > 5.78)”. However, there were so many voxels/vertices across the brain, the authors should perform some kind of multiple comparison correction for this number of voxels/vertices.

2. The authors concluded that GSCORR was not explained by physiological noise by simply compare the results with and without nuisance covariates regression. More methods like in Power et al. 2017 paper should be further tested.

3. The authors regressed out cardiac and respiratory signals. How did they calculate these two signals?

4. Reference style inconsistent. E.g., Line 54: “disorders like schizophrenia (14) and bipolar disorder (Zhang et al. 2018)”.

5. Line 386. Workbench not Workbranch.

6. Line 436. Please explain what theta is in this formula.

Power, J.D., Plitt, M., Laumann, T.O., Martin, A. (2017). Sources and implications of whole-brain fMRI signals in humans. Neuroimage, 146, 609-625, doi:10.1016/j.neuroimage.2016.09.038.

Liu, T.T., Nalci, A., Falahpour, M. (2017). The global signal in fMRI: Nuisance or Information? Neuroimage, 150, 213-229, doi:10.1016/j.neuroimage.2017.02.036.

Reviewer #2:

In this manuscript, Zhang and colleagues comprehensively examine properties of global fMRI signal in a large cohort of adults. They assess the stability of global signal across two resting-state runs, then the spatial topology modulated by tasks. Differences were observed and appropriately quantified by task state, and further analysis revealed that that these differences were likely due to changes in phase coherence (rather than amplitude variation). The authors further argue against a role for the basal forebrain (which I do not find convincing, see below). Overall, the work is rigorous, innovative, timely and compelling. 

2 minor revisions are recommended:

1) A very recent paper provides very complementary findings to the current work: Li et al (2019). Topography and behavioral relevance of the global signal in the human brain. Scientific Reports, 9, 14286. Although highly relevant, the Li paper does not diminish the novelty of the current report. In using the same sample, the global signal topology is again observed (not a true replication). However, the Li paper should be highlighted in the present paper for conveying the intrinsic structure of the global signal (i.e. network architecture underlying the signal) as well as the associations with behavior, thereby providing evidence that the global signal is meaningful. The current manuscript would benefit by including the Li paper in the intro, and relating the findings in the discussion to these observations.

2) I do not find the basal forebrain findings to be compelling. The authors are essentially interpreting the null hypothesis, which is problematic. The absence of a task > rest difference in basal forebrain contributions to global signal is not evidence for a lack of involvement. The basal forebrain is a very small structure with poor SNR, which may contribute to a Type II error. Further, the type of signals may not be sensitive to broad differences, but as possibly observable in different analytic frameworks, like representation similarity analysis. The authors must be far more conservative in their interpretation of the null effect of basal forebrain.

Reviewer #3:

The novelty and significance: The first exploration of an intrinsic architecture of global signal in the human brain.

Technical merit and the experimental design: HCP data is basing the technical advances on the most benefited neuroimaging methods. The whole experimental design is discovery and exploratory.

Sufficiency of the statistical analysis: There lacks a set of comprehensive statistical assessments on the global brain signal either for phase or amplitude. It seems a huge mount of spatial correlations done thought the whole analyses, however such analyses are problematic where the huge number of voxels (samples) are highly correlated in terms of their spatial adjacency. It is better to figure out a novel way to qualify the spatial similarity of the brain mapping measurements across different states of tasks and rests. One possibility would be Kendall's statistics (Jiang & Zuo, 2016, The Neuroscientist). It would be also another way to strengthen the statistical and translational significance by performing a test-retest reliability of the measurements on the global signal with the HCP test-retest data (please read Zuo et al., 2019, Nat Hum Behave for a reference).

All data needed to replicate the study: The HCP data is publicly shared to the community. Regarding the observations reported, the authors need to add some analyses of their reproducibility such as a replication with another public database (e.g., CoRR or INDI Lifespan Data).

---

## [Decision Letter · Decision Letter 2]

22 Apr 2020

Dear Dr Zhang,

Thank you for submitting your revised Research Article entitled "Rest-task Modulation of fMRI-derived Global Signal Topography is Mediated by Transient Co-activation Patterns" for publication in PLOS Biology. I have now obtained advice from one of the original reviewers and have discussed their comments with the Academic Editor. The Academic Editor has also provided some additional comments (see foot of this email).

Based on the reviews and the Academic Editor's comments, we will consider your manuscript further, assuming that you will modify the manuscript to address the remaining points raised. We will ask the Academic Editor to check whether s/he is satisfied with your responses. Please also make sure to address the data and other policy-related requests noted at the end of this email.

IMPORTANT:

a) Please address the remaining requests from reviewer #3.

b) The Academic Editor is puzzled as to why you corrected for multiple testing using FDR rather than the more conventional Bonferroni. Please perform a Bonferroni correction and report the results.

c) The Academic Editor notices a potential confound that may have serious consequences. As you will see, s/he feels that it would be unfair for you to fully address this issue at this late stage in the editorial process. However, you must mention this potential problem prominently so that the readers are made aware of it.

d) Please address my Data Policy requests below.

We expect to receive your revised manuscript within two weeks. Your revisions should address the specific points made by each reviewer. In addition to the remaining revisions and before we will be able to formally accept your manuscript and consider it "in press", we also need to ensure that your article conforms to our guidelines. A member of our team will be in touch shortly with a set of requests. As we can't proceed until these requirements are met, your swift response will help prevent delays to publication.

*Copyediting*

*Published Peer Review History*

*Early Version*

*Submitting Your Revision*

Sincerely,

Roli Roberts

Senior Editor

PLOS Biology

DATA POLICY:

Regardless of the method selected, please ensure that you provide the individual numerical values that underlie the summary data displayed in the following figure panels as they are essential for readers to assess your analysis and to reproduce it: specifically, many Fig panels display complex spatial pattern data (Figs 1B, 2BC, 3BC, 4B, 5BC, 6, 7, 8ABC, S1A, S2), which will need to be made available by one of the methods above; there are also simpler graphs (Figs 1C, 2AC 4B, 5CE, 8C, S1A, S2) for which we will need data (note that some Fig panels contain both, so are included in both lists). NOTE: the numerical data provided should include all replicates AND the way in which the plotted mean and errors were derived (it should not present only the mean/average values).

REVIEWERS' COMMENTS:

Reviewer #3:

[identifies himself as Xi-Nian Zuo]

Most issues I raised in the first round are addressed. I have a minor leaving for the next revision: While it is a good way of improving the spatial independence using the large-scale parcellation, the Pearson's corelation is not suitable for assessing the absolute spatial similarity, which should be evaluated by using intraclass correlation (ICC). Meanwhile, remember to report 95% CI (not only a single correlation value).

COMMENTS FROM THE ACADEMIC EDITOR [edited]:

The statistics are OK. It's a bit of a red flag though that the authors used False Discovery Rate which is a much more permissive way of correcting for multiple comparisons. Why not Bonferroni, which is the usual standard, and is much more stringent? Did they not get significance that way?

But there could be a bigger problem. I finally had time on my hands to just do a full review of the manuscript myself. And I think the paper has a serious conceptual problem that none of the reviewers dealt with.

The primary finding of the paper is that the GSCORR in the task condition has a significantly different distribution over cortex during task conditions than during the resting state; Moreover while the distribution over cortex matches respiration-evoked regressor in the resting state, the spatial distribution is different during tasks. 

However, consider the following null hypothesis. The GS is always the same under all circumstances whether task or rest state. During the task, the net response per voxel is a sum of the (fixed) GS + some task-evoked response. We can expect that the vector of the task-related response (over time, per specific voxel or brain region) will in general have some non-zero projection on the GS. There's no reason to expect that each task related response in each (gray matter) voxel will be precisely orthogonal to the GS over time. The amplitude of the dot product could be positive or negative. The GSCORR: correlation of the GS with the response per (gray matter) voxel, as defined by the authors in their Methods, would then be the sum of the fixed corr with the unchanging actual GS component for the voxel, plus a measure of the projection of the task-evoked response on the GS. If this value is negative the GSCORR for the voxel, as defined, would be smaller in magnitude than the resting state GSCORR. I can easily envisage situations where this would happen: our breathing (and heart rate) get strongly entrained to task timing when we're engaged in a task. The GS due to breathing (and HR) could thus easily have components that correlate or anticorrelate with the neurally driven task related response in a relevant brain area. This could lead to erroneous values for the GSCORR and an erroneous finding of a different distribution of GSCORR over cortex during the task.

At this late stage it is unfair to reject the ms or ask for major new analyses (I can't think of how to address the issue off the top of my head because the temporal patterns may be inextricably intertwined). But the authors should be required to bring up this possible confound in their discussion.

---

## [Editor Report · Decision Letter 3]

23 Jun 2020

Dear Dr Zhang,

On behalf of my colleagues and the Academic Editor, Aniruddha Das, I am pleased to inform you that we will be delighted to publish your Research Article in PLOS Biology. 

Early Version

PRESS 

Kind regards,

Alice Musson

Publishing Editor, 

PLOS Biology

on behalf of

Roland Roberts,

Senior Editor

PLOS Biology